# Geographical distribution of freshwater fishes in Saudi Arabia

**Ibrahim G. Alharthi**[1,2]*, **I. G. Cowx**[1], **Jon P. Harvey**[1]

1 Hull international Fisheries Institute, University of Hull, Kingston upon Hull, United Kingdom, 2 National Centre for Wildlife, Riyadh, Saudi Arabia

* i.alharthi@ncw.gov.sa

**Data Availability Statement:** All relevant data are within the manuscript and its Supporting Information files.

**Funding:** National Centre for Wildlife for providing PhD scholarship to Ibrahim G. Alharthi including

## Abstract

Species presence/absence data in different water bodies in different regions of Saudi Arabia were collated from the literature and collected from field surveys to determine the geographical distribution of fish species in the country. Freshwater fish are mainly located in drainages in the south-west of the Kingdom of Saudi Arabia, both in the lowlands (western drainage systems) and highlands (eastern drainage systems) of the Sarawat Mountain Range. The eastern drainage systems were dominated by three endemic species, while the western drainage systems had a variety of endemic and non-native species. Ten non-native fish species were reported, mainly in artificial water bodies in the north and east of the country, but also in dams located in Al Baha Region, Abha and Rabigh, meaning both western and eastern drainage systems are being colonised by non-native species such as *Oreochromis* and *Carassius* species.

## Introduction

Understanding the distribution and basic ecology of aquatic biota, and the factors driving those ecological characteristics, are essential to underpin management plans for the conservation of biodiversity and associated ecosystems [1]. It is essential to understand the underlying reasons for the distribution of fish species and whether this is driven by biogeographical factors, changes in exploitation pressures, degradation and change in ecosystem quality and functioning [2]. Of particularly importance is an understanding of any causes for the disappearance of species from a particular location so remedial and restorative measures can be formulated to conserve biodiversity.

One geographic area characterized by a lack of comprehensive understanding regarding the distribution and ecology of aquatic biodiversity is the Arabian Peninsula. Freshwater fishes in Saudi Arabia, for example, are a vital part of the country's natural heritage and provide ecological, social, and economic benefits, but information on these bioresources is limited mostly to species inventories [3–8]. These inventories have been updated periodically [9–16], but are restricted in the species covered, and largely target endemic fauna [17]. It is crucial that information on the distribution of species is current and includes all species as well as the challenges conserving native fish species.

financial support for the study. The funders had no role in study design, data collection and analysis, decision to publish, or preparation of the manuscript.

**Competing interests:** The authors have declared that no competing interests exist.

Two main natural drainage systems are found in Saudi Arabia: the eastern and western drainage basins (Fig 1). The drainage systems are separated by the Mountain Range that extends along the western Saudi Arabia with a range of elevations between 1600 and 3000 m above sea level (Fig 1). These drainage systems have the greatest availability of surface water and are the main areas inhabited by freshwater fishes in Saudi Arabia. There are also freshwater bodies in the Central Region and the Eastern Province, but only one species of secondary freshwater fish occurs there.

The eastern drainage system flows towards the east or north-east of Saudi Arabia. Krupp termed these the Empty Quarter drainage [5]. These eastern drainage systems end in an endorheic basins. The riverbed, or wadis, as they are known regionally, are seasonal and/or intermittent water bodies and have little flow because of the dry conditions. They serve mostly to connect permanent water bodies in times of rainfall. The eastern system is distinguished by six main basins: Wadi Najran Basin; Wadi Habawnah Basin; Wadi Tathleeth Basin; Wadi Bishah Basin; Wadi Ranyah Basin and Wadi Turabah Basin (Fig 1). The latter three wadis have larger supplies of water because of higher rainfall in the southern region.

The western drainage systems (Red Sea drainages) flow west towards the Red Sea and some run parallel to coastline (Fig 1). The wadis are divided into four groups based on their location and discharge. The first group (G1) includes basins located between the Yemen border in the south to the Wadi Heli basin in the north (Fig 1). There are 24 wadis within an estimated area of 29,250 km$^2$, and these wadis discharge towards the Red Sea over a shoreline length of 340 km. Wadis in G1 are characterized by high levels of water discharge and are prone to flooding. The second group of wadies (G2) originate between Wadi Heli and the Wadi Al-Ahsabah basin in the north. Flows in G2 wadis occur during the same period as those in G1 wadis, but they are less intense, do not occur in all years and are considered less disruptive to people and infrastructure. The third group (G3) comprises drainages between Wadi Al-Ahsabah and Wadi Sa'dya, over a coastline length of approximately 180 km. Wadis in this area rarely flood and only for short periods, and they do not reach the Red Sea coast. The fourth group (G4) comprises drainages between Wadi Sa'dya and Umluj city. Freshwater fish are reported from few locations in this area, possibly indicating that wadis in this group do not support as many permanent streams and pools that can be inhabited by freshwater fishes. The relatively high discharge in the southern wadis is due to the monsoonal regime in that part of Saudi Arabia.

Other areas with wadis and freshwater bodies are the central Al-Madinah Province (M), including Khyber, Khadra and Wadi Hadiyah where freshwater fish are present. The main wadi channels in the Al-Madinah are Wadi Al-Hemad or Idham. Its length is approximately 400 km. It starts from Al-Madinah Province until it empties into the Red Sea, 107 km north of Umluj city (Fig 1).

This information is critical because main of the main drainage basins inhabited by endemic freshwater fishes are also attractive for development of water infrastructure as they are sources of abundant surface water, and many have been impounded for domestic supply and agriculture. Some 70% of the dams (about 357 of 500) in Saudi Arabia are in regions occupied by endemic freshwater fish. There are many dams with large storage capacities, especially in Asir (171 dams), Makkah (57 dams) and Al Baha (48 dams) regions, which overlap with areas occupied by endemic freshwater fishes. These could be a major issue for the sustainability of these species because the dams act as barriers to fish migration, disrupt connectivity and cause fragmentation of populations. Whilst the impounded areas may act as refugia they also inundate valuable spawning habitat for riverine species.

This study provides a detailed update on the distribution of Saudi Arabian freshwater fishes based on existing literature and field sampling. Information on the main factors influencing the distribution of these species was also collected to understand the potential pressures acting

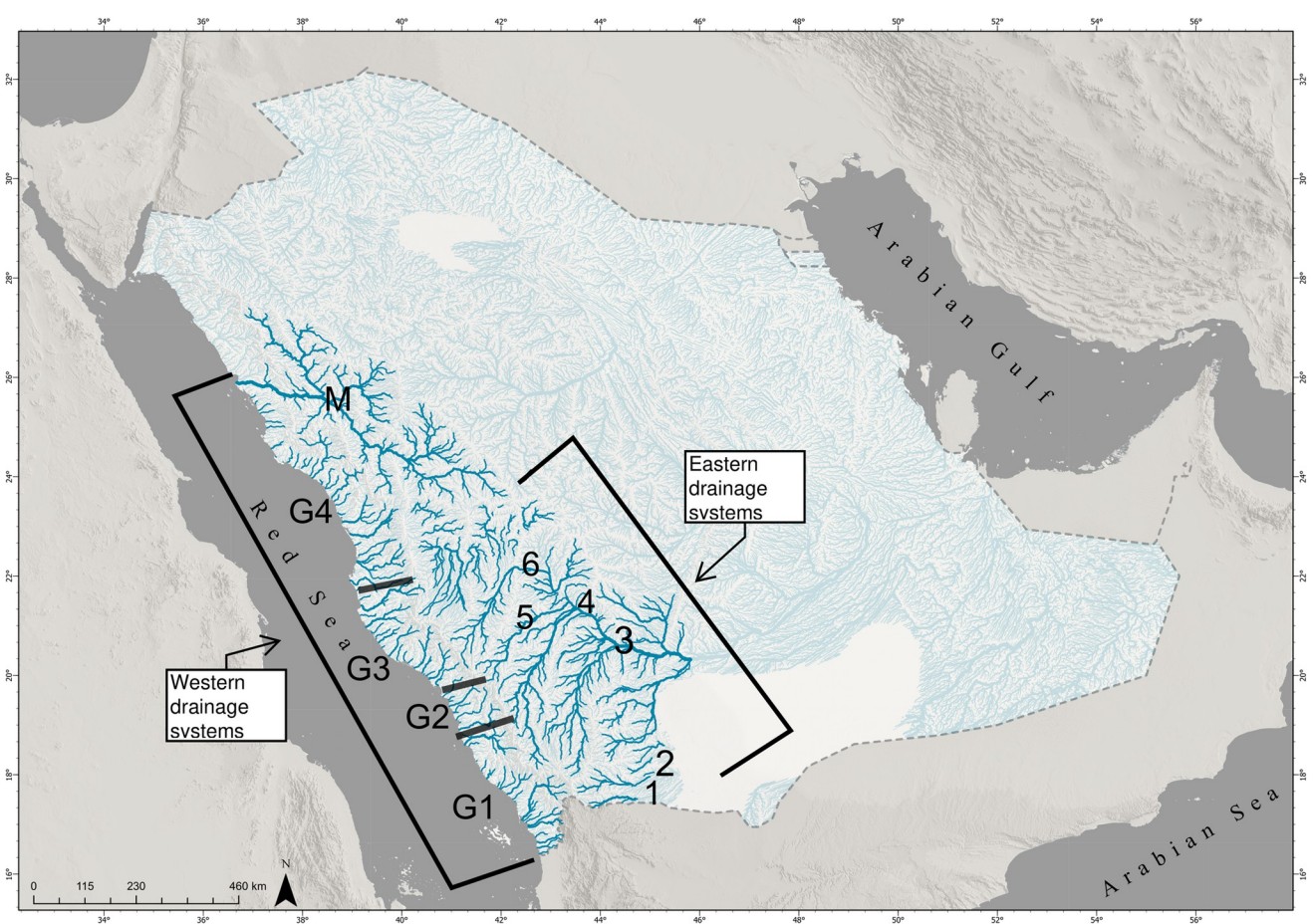

**Fig 1. Location of the main eastern and western river basins in Saudi Arabia where indigenous primary freshwater fishes occur.** The eastern basin that drain towards the east or north-east of Saudi Arabia: 1 - Wadi Najran, 2 - Wadi Habawnah Basin, 3 - Wadi Tathleeth Basin, 4 - Wadi Bishah Basin, 5 -Wadi Ranyah Basin, 6 - Wadi Turabah Basin in the highland. The western basin that drain towards the Red sea: G1, G2, G3, G4, M (Al-Madinah Province) major wadis groups within southwest and northwest lowlands. (Contains information from OpenStreetMap and OpenStreetMap Foundation, which is made available under the Open Database License).

on various populations and ecosystems, including changes to, or degradation of, habitats. Furthermore, the distribution of non-native fish species is mapped; this is considered critical because little is known about the spread and invasiveness of non-native fish species in Saudi Arabia [18, 19]. The information is used to prioritize the most important water basins and wadis (river beds) that will be the focus of future conservation management plans in the country.

## Materials and methods

The distribution of freshwater fish species in Saudi Arabia was assessed by collating geographical coordinates of locations where fish species were reported in the literature [5–15, 19] and 16 sites sampled in this project (see Supplementary S1 Table for locations of water bodies). The analysis was based on eight indigenous freshwater fish species, 10 introduced species and two secondary species. The latter are species that have entered inland water habitats from the Red Sea or Arabian Gulf and are able to tolerate a wide range of saline conditions. The open-source GIS (Geographic Information System) database of the Ministry of Environment, Water and

Agriculture (mewa.gov.sa) was examined for dam locations and available information about major drainages and wadi systems. The geographical coordinators were then plotted using IUCN Freshwater Species Mapping Standards (www.iucnredlist.org/resources/mappingstandards) was used to generate maps containing the distribution records of different fish species according to the time the studies were conducted. Distribution maps were also produced for non-native fish species.

Multivariate analysis, based on species presence in different water bodies in different regions, was used to characterise geographical distribution of freshwater fish species in Saudi Arabia. Similarity of fish species assemblages in each water body was compared using the Jaccard index based on presence-absence of different species. The data were submitted to a similarity matrix and cluster analysis was carried out to group sites with similar species assemblages. Data were also ordinated using non-metric multidimensional scaling (nMDS) to investigate similarities in the species composition in different localities and the main species contributing to the similarities was determined using the SIMPER tool. The matrices were then submitted to permutational multivariate analysis of variance (PERMANOVA) (9999 random permutations) to assess the average similarity of species composition between sites based on the Jaccard resemblance matrix [20–22]. All analyses were carried out in PRIMER 6 (Plymouth Routines in Multivariate Ecological Research).

All procedures were carried out in accordance with the University of Hull, UK, ethical procedures committee for care and use of animals in research. The specimens have not been permanently deposited in a scientific collection.

## Results

### Geographical distribution of freshwater fishes in Saudi Arabia

The distribution of indigenous primary and secondary freshwater fish species, as well as non-native fish species, is illustrated using maps provided in the supplementary material. Fish species present in the Empty Quarter eastern drainages are *Carasobarbus apoensis* (Banister & Clarke), *Cyprinion mhalense* Alkahem & Behnke and *Garra buettikeri* Krupp. The dominant native species in the western, lowland drainages discharging towards the Red Sea are *Garra tibanica* Trewavas, *Garra sahilia* Krupp, *Cyprinion acinaces* (Banister & Clarke), *Arabibarbus arabicus* (Trewavas), *Aphaniops dispar* (Rüppell), although Borkenhagen & Krupp [11] mentioned that *C. apoensis* exists in both eastern and western drainages. Secondary freshwater fish species include *Aphaniops stoliczkanus* and *Aphaniops dispar*. Non-native fish species, especially *Oreochromis* and *Poecilia* species, which have been introduced for aquaculture and vector control, occur mostly in the central and eastern regions of the Kingdom, although some species such as *Carassius auratus* and *Oreochromis* spp. have been introduced into dams in the south-western region where endemic species typically exist.

This distribution of freshwater fish species assemblages in Saudi Arabia was further categorised into six regions using Multidimensional Scaling of species presence/absence (Fig 2), viz: eastern drainage systems located in highlands in the south and south-west of the Kingdom; western drainage systems in the coastal plains in the south and south-west region; Eastern Province; Central Region; North Central Region; and Northern Borders Province. The species contributing to these groupings of sites were as follows.

Three-endemic species *Carasobarbus apoensis*, *Cyprinion mhalense* and *G. buettikeri* were predominant in the eastern drainage systems contributing 48.7% similarity between wadis and dams in this system. *Carassius* spp. and *Oreochromis* spp. (non-native species) were also caught during trips associated with this study.

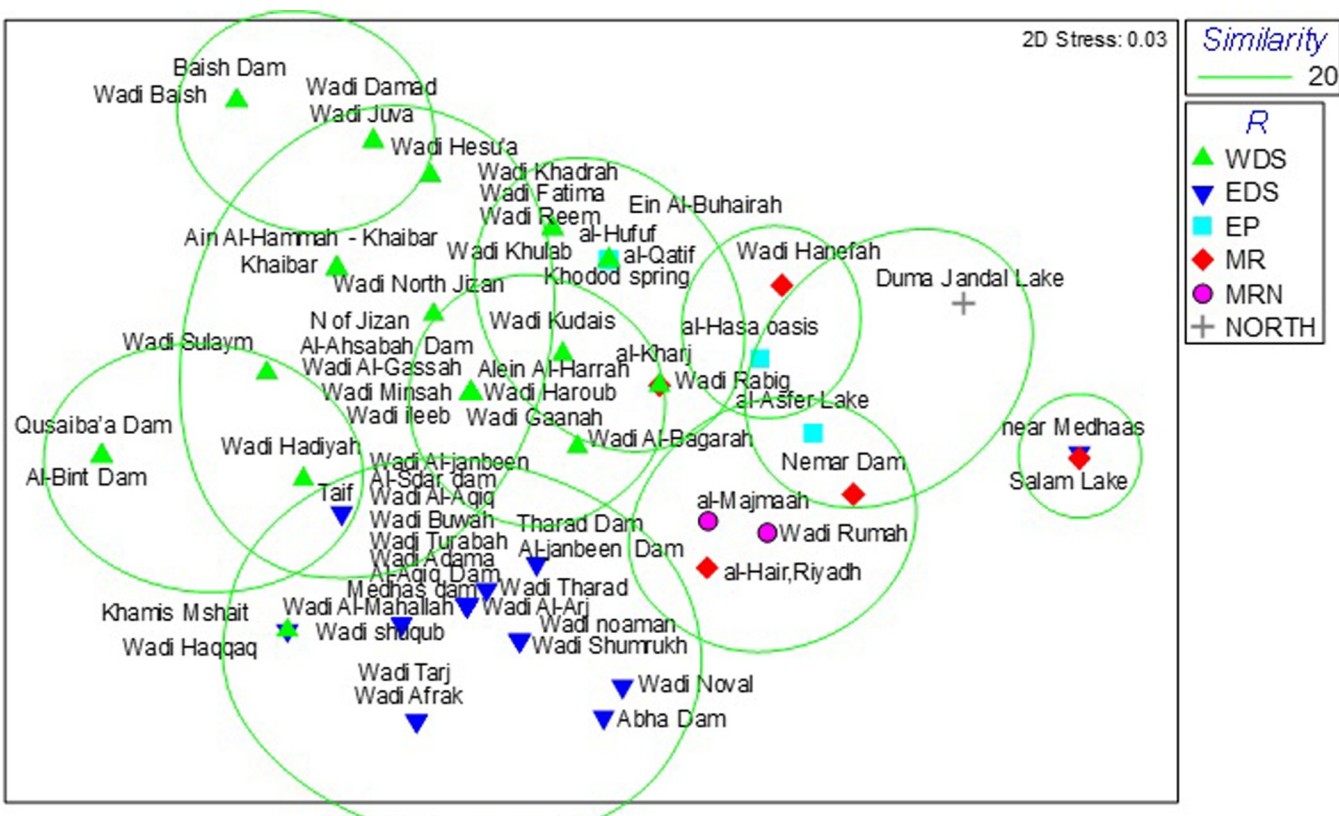

**Fig 2. Non-metric multidimensional scaling (nMDS) ordination plot to compare species presence/absence similarity between all location records within different drainage systems regions.** WDS = Western drainage systems; EDS = Eastern drainage systems; EP = Eastern Province; MR = Central Region; MRN = North Central; NORTH = Northern Borders Province Region. Clusters grouped using SIMPER analysis. For overlapping names see supplementary S1 Table - locations of water bodies.

*Garra tibanica*, *G. sahilia*, *Cyprinion acinaces*, *Arabibarbus arabicus*, *Acanthobrama hadiya-hensis* and *A. dispar* are the predominant species in the western drainage systems. *Garra sahilia*, *A. dispar*, *G. tibanica* and *A. arabicus* contributed 17.6% of species similarity in this system. There are also several sub-groups of wadis in these systems, with the presence of several endemic species, such as *A. arabicus* and *A. hadiyahensis*, contributing to the dissimilarity. *Acanthobrama hadiyahensis* is known only from a few water bodies in the far north of the western drainage systems, including Wadi Hadiyah, Al-Bint Dam and Qusaiba'a Dam, and *A. arabicus* occurs in the south of the western drainage systems and into Yemen water bodies. *Aphaniops dispar* is a secondary freshwater fish occurring in the western drainages, while *A. stoliczkanus* is known from the Central Region and the Eastern Province. The non-native species, *Oreochromis* spp., are also becoming more prominent in some wadis in the western drainage system.

Non-native species (*Oreochromis* spp., *Gambusia holbrooki* Girard, *Poecilia latipinna* (Lesueur), *Poecilia reticulata* Peters, *Xiphiphorus maculatus* (Günther), *Clarias gariepinus* (Burchell), *Ctenopharyngodon idella* (Valenciennes), *Carassius carassius* (L.) and *Carassius auratus* (L.)) dominate in artificial water bodies, such as ponds, lakes and irrigation channels, in the Central and Northern regions and Eastern Province, and account for between 10% and 50% of the similarity in fish assemblages between sites. Dry wadis into which urban wastewater

is discharged and artificial lakes constructed to receive wastewater are often inhabited by non-native fish species that can withstand stressful environmental conditions.

Several sub-clusters of water bodies were discriminated in the different regions based on species presence using hierarchical cluster analysis (Fig 3). The first comprised Baish Dam, Wadi Baish, Wadi Damad, Wadi Juva, Taif, Wadi Sulaym, Ain Al-Hammah–Khaibar, Khaibar, Wadi Hesu'a and Wadi North Jizan in WDS and EDS regions, which shared four indigenous fish species: *A. arabicus*, *C. acinaces*, *C. mhalense* and *G. tibanica*. The second cluster in WDS region was discriminated by two indigenous fish species (*G. buettikeri*, *G. sahilia*) and one secondary freshwater fish species (*A. dispar*), and comprised Alein Al-Harrah, Al-Ahsabah Dam, N of Jizan, Wadi Al-Gassah, Wadi Gaanah, Wadi Haroub, Wadi ileeb, Wadi Minsah, Wadi Al-Bagarah and Wadi Kudais. Hamidan & Shobrak (2019) found *G. buettikeri* in Wadi Al-Bagarah, which is a lowland drainage system, but as this species usually occupies highland drainage systems, further investigation is needed, especially because this species has similar features with *Garra tibanica* that inhabits lowland drainage systems. The third cluster in EP, North, MR and MRN regions comprised al-Asfer Lake, Dumat Al-Jandal, al-Hair-Riyadh, al-Majmaah, Nemar Dam and Wadi Rumah, and shared six non-native fish species, namely *Oreochromis niloticus* (L.), *Oreochromis aureus* (Steindachner), *G. holbrooki*, *P. latipinna*, *C. gariepinus* and *C. auratus*. Wadi Hanefah and al-Hasa oasis in EP and MR regions accounted for a fourth cluster and shared six introduced fish species (*O. niloticus*, *G. holbrooki*, *P. latipinna*, *X. maculatus*, *C. gariepinus*, *C. idella*), one secondary freshwater fish (*A. dispar*) and one land-locked fish species (*M. cephalus*). A fifth cluster grouping al-Kharj, Wadi Rabig, Wadi Fatima, Wadi Khadrah, Wadi Khulab, Wadi Reem, al-Hufuf, al-Qatif, Ein Al-Buhairah and Khodod spring in WDS and EP regions was discriminated by the presence of one indigenous fish species (*G. tibanica*), one secondary freshwater fish species (*A. stoliczkanus*) and one introduced fish species (*O. niloticus*). Three dominant endemic fish species, *C. apoensis*, *C. mhalense* and *G. buettikeri*, accounted for a sixth cluster comprising Abha Dam, Wadi Noval, Khamis Mshait, Wadi Haqqaq, Wadi Afrak, Wadi Tarj, Wadi Shumrukh, Wadi Al-Arj, Wadi noaman, Wadi Al-Mahallah, Wadi shuqub, Thrrad Dam, Al-Janbeen Dam, Wadi Turabah, Al-Aqiq Dam, AL-Sdar Dam, Wadi Buwah, Wadi Thrrad, Medhas Dam, Wadi Adama, Wadi Al-Aqiq and Wadi Al-Janbeen in EDS and WDS regions. The presence of *Carasobarbus*

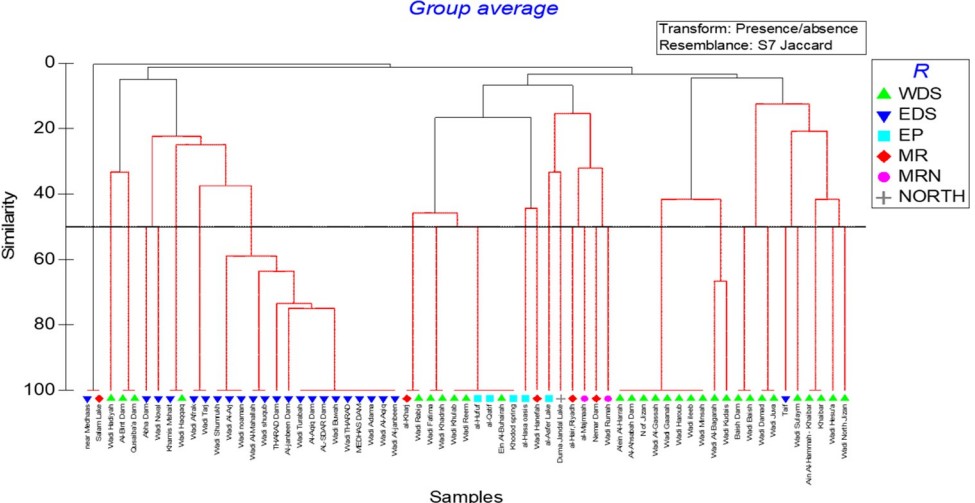

**Fig 3. Hierarchical cluster analysis to compare species presence/absence similarity which fish members are shared and which are separate in all location records.**

*apoensis*, *A. hadiyahensis*, *C. acinaces* accounted for a small cluster in WDS region, comprising Wadi Hadiyah, Al-Bint Dam and Qusaiba'a Dam, and the non-native species *C, auratus* grouped sites near Medhaas and Salam Lake in EDS and MR region.

## Distribution of non-native freshwater fish species in Saudi Arabia

At least 20 non-native freshwater fish species have been recorded as being introduced into Saudi Arabia [23], but only ten species are reported to have established in the wild, mostly in two regions: the central Riyadh Region and Eastern Province. Some non-native fish species are also found in areas overlapping the natural ranges of indigenous fish species in the eastern Empty Quarter drainages and the western, lowland drainages discharging towards the Red Sea (Fig 4).

*Oreochromis niloticus* is found in wadis and man-made lakes in Central Region, including Wadi Haneefah, Nemar Dam, Al-Hair, Al-Majmaah and Al-Kharj, as well as in Al-Hasa canal and agricultural waste drainage lakes, such as Al-Asfer Lake and Al-Wyoon Lake in Eastern Province (Fig 4). The species was also recorded in Thrrad Dam. Other tilapiine cichlids found include *O, aureus* in Wadi Al-Hair wetland in Central Region and *Oreochromis spilurus*

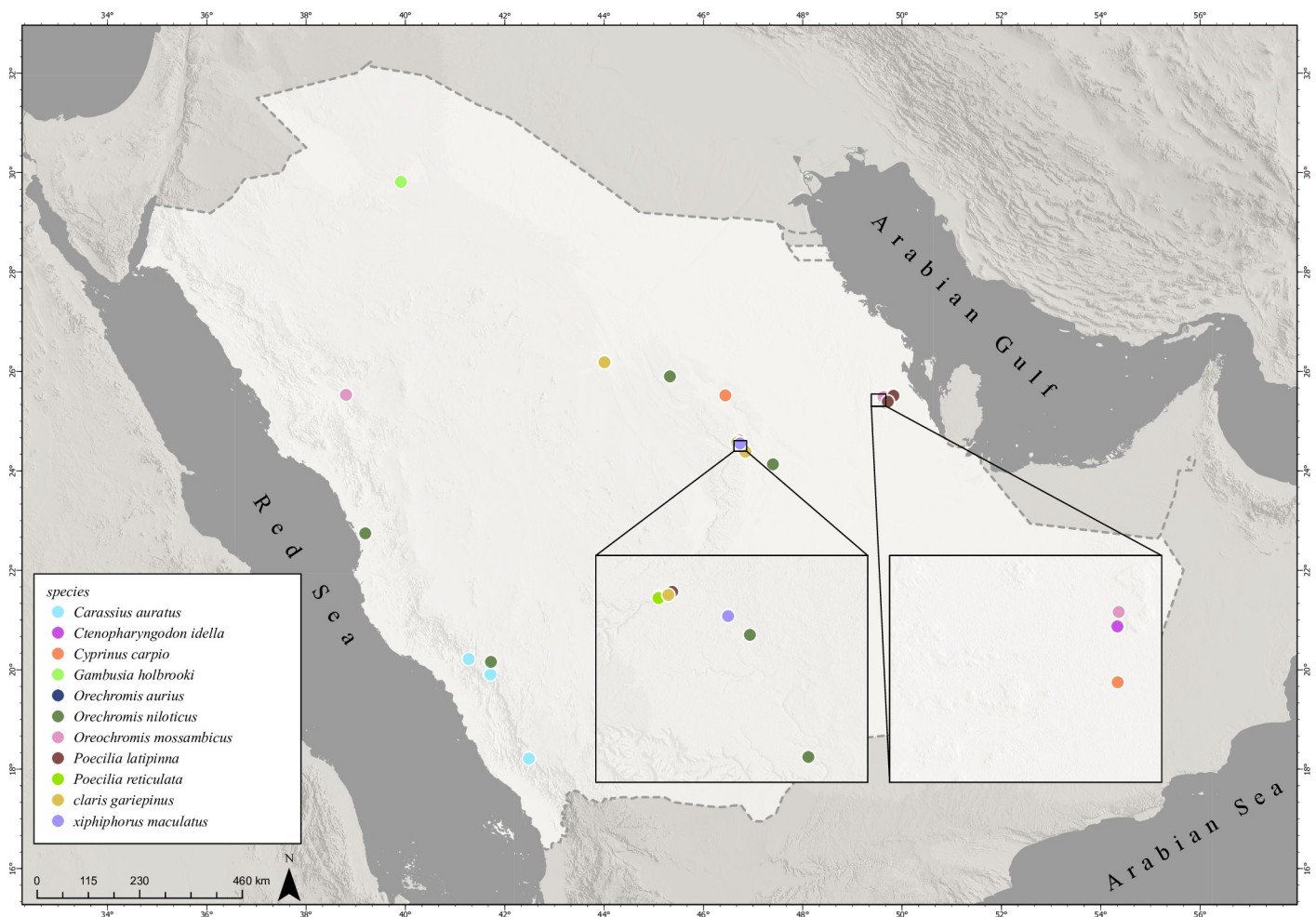

**Fig 4. Distribution of non-native freshwater fish species in Saudi Arabia.** Contains information from OpenStreetMap and OpenStreetMap Foundation, which is made available under the Open Database License.

(Günther), which is widely cultured in floating cages in the Red Sea and has been stocked in artificial lakes in the Northern Borders Region (Fig 4).

*Gambusia holbrooki* is found in many areas including Al-Asfer Lake and channels in Al-Hasa (Fig 4). This species was released into Dumat Al-Jandal Lake in 2018 together with tilapia and sea bream. *Poecilia latipinna* is found in the Eastern Province in Al-Hofuf, Al-Qatif and Al-Ahsa oases within agricultural irrigation channels or man-made lakes, including Al-Asfer Lake (Fig 4). This species also occurs in the Central region in Riyadh urban discharge wadis and artificial wetlands, such as Nemar Dam (Fig 4). *Poecilia reticulata* and *X. maculatus* are found in Wadi Haneefah and in springs and irrigation channels in Eastern Province (Fig 4). *Clarias gariepinus* is found in urban drainage wadis in Riyadh, such as Wadi Haneefah, Nemar Dam and Al-Hair and in Wadi Rumah in Al Qassim Region (Fig 4). *Ctenopharyngodon idella* has been introduced into irrigation channels in Al-Hasa Oasis, while both *C. carassius* and *C. auratus* (L.) have been introduced into Saudi for ornamental purposes, but only *C. carassius* has been reported in the wild in Al-Janabeen Dam (Fig 4).

## Discussion

A total of 23 freshwater fish species have been identified in Saudi Arabia [16]. Of these species three are classified as threatened (*C. apoensis* [Endangered], *A. hadiyahensis* [Critically Endangered] and *G. buettikeri* [Vulnerable]) according to the IUCN Red List, and two (*C. mhalense* and *G. tibanica*) are in decline in population size [24].

Freshwater fishes in Saudi Arabia are mainly located within drainages in the south-west of the country, both in the lowlands (western drainage systems) and highlands (eastern drainage systems) of the Sarawat Mountain Range. Three endemic species (*C. apoensis*, *C. mhalense* and *G. buettikeri*) dominate the eastern drainage systems. *Garra tibanica*, *G. sahilia*, *C. acinaces*, *A. arabicus*, *A. hadiyahensis* and *A. dispar* occur in the western drainage systems. The distribution of the indigenous freshwater fish fauna appears to be driven by paleogeographic events and physical and ecological conditions between wadis and drainages. Differences in environmental conditions, especially flow regimes, are also more obvious between in water bodies in the western drainage system. Non-native fish species have become a common feature of the freshwater fish fauna in Saudi Arabia. They have been introduced into Eastern Province, Central Region, north Central and Northern Borders Province for biological control (controlling algal blooms, aquatic plants, snails, mosquitos), ornamental fishes, aquaculture and unintentionally [23, 25, 26], and some artificial lakes have been stocked for recreational fishing. *Poecilia latipinna*, *P. reticulata*, *X. maculatus*, *C. gariepinus* and *Oreochromis* spp. are common in the Riyadh Region, including in Wadi Haneefah and its tributaries in Riyadh City, plus other waterways and springs near Al-Kharj City. These include temporary waters, such as rawdat (= meadow), marshlands and fayadh (flooded areas), which are created during the rainy season or flooding period, and have been invaded by fish escaping from nearby fish farms. They are also found in pools created by discharge of waste water into dry wadis, which creates aquatic habitat that can be occupied by these non-native species that are able to tolerate poor water quality conditions.

In the Eastern Province, several cities, such as Al-Ahsa, Al-Qatif and Al-Hofuf, are characterised by springs, oases and irrigation channels for agriculture, especially palm trees, and these channels usually discharge into artificial lakes. Water used for irrigation has become semi-saline and salinity rates are increasing. Species that can tolerate a wide range of salinity have been introduced for biological control of mosquitoes or weeds. In addition, *M. cephalus* and *A. stoliczkanus*, which are native to Saudi Arabia, have ingressed into the channels to exploit the changing environment. However, *A. stoliczkanus* populations are generally in decline because of interaction with non-native species and habitat degradation [9].

Importantly, this study recorded the presence of non-native fish species in dams located in Al Baha Region, Abha and Rabigh wadis, which means both western and eastern drainage systems are being colonised by non-native species, including *Oreochromis* and *Carassius* species. The arrival of these non-native fish species to this part of Saudi Arabia highlights the potential threat to local endemic fish species, and the need for more surveys of all dams and wadis in the region to understand better their overall status and potential impacts.

Apart from the introduction and dispersion of non-native species, the freshwater fish fauna of Saudi Arabia is threatened by the construction of dams, protracted drought periods as well as regulation of flows and abstraction of water for drinking or agriculture [17]. Also of concern is the recolonisation of wadis after periods of low or no rainfall, including drought periods. It appears that fish survive dry and low flow conditions in refuge pools and habitats in the upper reaches of wadis until the rains return, and then fish colonise the wadis in a downstream direction with the flood waters. For example, adult fish survive dry periods in small pools with no flowing water connections in the upper reaches of Wadi Turabah close to its headwaters in the Buthrah Mountain. These pools act as refugia until the rainy season and then fish disperse with the flood to recolonise the lower reaches. The importance of such refugia is common in intermittent streams in drought prone areas and was highlighted by Pires et al. [27] as essential habitats to protect organisms from the impact of droughts. This survival strategy, where small pools and streams in the upper reaches of wadis or potentially wetlands on floodplains act as refugia, needs further investigation. There is also a need to understand how fish behave to avoid the extreme hydrological and hydraulic conditions during flood events, which can displace adult, juvenile and egg stages, and cause in large-scale movements of sediments and debris. This is particularly important because some fish species potentially migrate upstream to the headwaters of wadis when there is sufficient flow and the ecological characteristics of this migration strategy need further investigation, particularly in relation to recruitment dynamics.

Dams have exacerbated this problem of species natural dispersal during high flow events. For example, many fishes have been observed near the gates of Al Sadr Dam after releasing water during a flood event (pers. obs.). Dams also regulate flows and may cause desertification and more intense drought conditions in areas that were traditionally refuge areas, but these habitats are now disappearing due to insufficient flow being released from the dams and high levels of abstraction of water. The impact of these changes in flow characteristics on fish and fisheries needs to be fully understood in Saudi Arabia to enable flows to be optimized for the protection of native species. Although information on the biogeography and autecology of indigenous freshwater fish species in Saudi Arabia is limited, it is clear that the fish assemblages are under considerable pressure from an array of threats, especially construction of dams, flow regulation and water abstraction [28].

This study has highlighted the most important drainages and wadis in Saudi Arabia, especially in the eastern and western drainage systems [29], that need to be targeted and managed to protect freshwater fish biodiversity of the Kingdom. The distribution of dams, and regulation of flows and abstraction of water for drinking or agriculture, will likely have profound impacts on endemic aquatic biodiversity in the Kingdom and needs effective planning and management. The information provided in this paper should be used in the formulation of fisheries management plans in Saudi Arabia to rationalise development and management of dam operations in the key areas where endemic fish are present. This linkage between location and operation of dams and water abstraction infrastructure, and distribution of fishes is critical to formulate appropriate management plans to protect and conserve these endemic species and is a key step to ensure measures are taken to minimise or mitigate any impacts on freshwater biodiversity. In addition, the role of fish refuges during drought periods needs further

research and fish refuges need protection in conservation management plans because they are key habitats that are fundamental to the sustainability of fish populations in these drought prone systems in Saudi Arabia.

## Supporting information

**S1 Table. The location fish species and the water bodies in Saudi Arabia.**
(DOCX)

**S1 File. Distribution maps of all the study species.** Contains information from OpenStreet-Map and OpenStreetMap Foundation, which is made available under the Open Database License.
(RAR)

## Acknowledgments

We are grateful for the support and assistance of numerous individuals and organizations in the completion of this study. I would like to thank the National Centre for Wildlife of (the Kingdom of Saudi Arabia) for permission to conduct this research, special thanks to Mr. Muhammad Al-Nashiri from GIS department for his invaluable contribution in enriching and improving the quality of maps. Thanks are extended to the University of Hull International Fisheries Institute staff for supporting my research.

## Author Contributions

**Conceptualization:** Ibrahim G. Alharthi, I. G. Cowx, Jon P. Harvey.

**Data curation:** Ibrahim G. Alharthi, I. G. Cowx, Jon P. Harvey.

**Formal analysis:** Ibrahim G. Alharthi.

**Investigation:** Ibrahim G. Alharthi, I. G. Cowx, Jon P. Harvey.

**Methodology:** Ibrahim G. Alharthi, I. G. Cowx, Jon P. Harvey.

**Project administration:** Ibrahim G. Alharthi.

**Software:** Ibrahim G. Alharthi.

**Supervision:** I. G. Cowx, Jon P. Harvey.

**Validation:** Ibrahim G. Alharthi.

**Writing – original draft:** Ibrahim G. Alharthi, I. G. Cowx, Jon P. Harvey.

**Writing – review & editing:** Ibrahim G. Alharthi, I. G. Cowx, Jon P. Harvey.

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
