## [Decision Letter · Decision Letter 0]

10 May 2024

PONE-D-24-14979Biogeography of freshwater fishes in Saudi ArabiaPLOS ONE

Dear Dr. alharthi,

Thank you for submitting your manuscript to PLOS ONE. After careful consideration, we feel that it has merit but does not fully meet PLOS ONE’s publication criteria as it currently stands. Therefore, we invite you to submit a revised version of the manuscript that addresses the points raised during the review process.

We look forward to receiving your revised manuscript.

Kind regards,

Benigno Elvira, Ph.D.

Academic Editor

PLOS ONE

Journal Requirements:

"National Centre for Wildlife for providing PhD scholarship to Ibrahim G. Alharthi including financial support for the study."

"We are grateful for the support and assistance of numerous individuals and organizations in the completion of this study. I would like to thank the National Centre for Wildlife of (the Kingdom of Saudi Arabia for permission to conduct this research and for providing the necessary resources and facilities, special thanks to Mr. Muhammad Al-Nashiri from GIS department for his invaluable contribution in enriching and improving the quality of maps. Thanks are extended to the University of Hull International Fisheries Institute staff for supporting my research."

"National Centre for Wildlife for providing PhD scholarship to Ibrahim G. Alharthi including financial support for the study."

4. We note that Figures 1, 2, 5 and Supporting Information in your submission contain map/satellite images which may be copyrighted. All PLOS content is published under the Creative Commons Attribution License (CC BY 4.0), which means that the manuscript, images, and Supporting Information files will be freely available online, and any third party is permitted to access, download, copy, distribute, and use these materials in any way, even commercially, with proper attribution. For these reasons, we cannot publish previously copyrighted maps or satellite images created using proprietary data, such as Google software (Google Maps, Street View, and Earth). For more information, see our copyright guidelines: http://journals.plos.org/plosone/s/licenses-and-copyright.

a. You may seek permission from the original copyright holder of Figures 1, 2, 5 and Supporting Information to publish the content specifically under the CC BY 4.0 license.  

Reviewers' comments:

Reviewer's Responses to Questions

**Comments to the Author**

1. Is the manuscript technically sound, and do the data support the conclusions?

Reviewer #1: Yes

Reviewer #2: Partly

2. Has the statistical analysis been performed appropriately and rigorously? 

Reviewer #1: Yes

Reviewer #2: Yes

3. Have the authors made all data underlying the findings in their manuscript fully available?

Reviewer #1: No

Reviewer #2: No

4. Is the manuscript presented in an intelligible fashion and written in standard English?

Reviewer #1: Yes

Reviewer #2: Yes

5. Review Comments to the Author

Reviewer #1: This is a nice manuscript and I see no major obstracles to publush this. Indeed, old data had been compiled for that book - https://www.researchgate.net/publication/341322582_Freshwater_Fishes_of_the_Arabian_Peninsula ignored in the manuscript, and replaced by a checklist, which just copies the book/steals the information. Maybe not nice.

I have two points - I do not believe that Carassius carassius was introduced and your records must be based on misidentification - please check again and write a short sentece why - and show a picture of such fish. The identification of C. gibelio is likely also wrong - and all are likely to be C. auratus. Please explain some things. Forthermore, I like to see some text how you identified the Oreochromis - this is not trivial and maybe here also you show some pictures. Last but not lease, please add a table to the suppl. materials in which you make clear, which species you have found at which place - this is only for your own work. Basically an excel with a line for each record: species/coordinates/date..... is the way. By this you make your new records available

Thans and good luck

Reviewer #2: The authors document and analyse the distribution of inland water fishes in Saudi Arabia, based on literature records and field surveys, adding new and unpublished data. The paper will be of great importance for inland water fish conservation efforts in the Kingdom of Saudi Arabia and deserves to be published. Here below, I am offering extensive comments, which I trust will help the authors to revise the paper.

My major concerns with the current draft of the paper are as follows: The title reads “Biogeography of freshwater fishes in Saudi Arabia”. Biogeography is defined as “the study of the distribution of organisms across geographic space, and the processes that shape these distributions”. This paper reports the geographic distribution, but does not analyse the processes that shape these distributions. A key factor in shaping the distribution of native inland water fishes is the geological history of the drainage basins, which is not included in this paper. For this reason I would not consider this a biogeographic study. Biogeographic studies only make sense for native species.

The paper would benefit from a clearer distinction between native primary and secondary freshwater fishes and non-native species, above all when reporting similarities in fish populations among drainage systems. There needs to be an explanation why native and non-native species are combined in the cluster analysis. This might generate results that are useful for conservation, but the approach does not provide biogeographic information. The inclusion of historic records and results of recent surveys needs further explanation and discussion.

Here below are more detailed comments. I have also mentioned typos and errors in writing style, because PLOS ONE does not provide copy editing.

The Abstract is not informative. It should summarize major findings and conclusions.

Introduction: An important reference is missing: “Freyhof, J., Els, J. Feulner, G.R., Hamidan, N.A. & Krupp, F. (2020). The Freshwater Fishes of the Arabian Peninsula. Dubai, Motivate Publishing. 272 pp.” This monograph includes the most recently published distribution maps of freshwater fishes native to the Arabian Peninsula.

Another paper that should be added is: Ross, W. (1985). Oasis fishes of Eastern Saudi Arabia. Fauna of Saudi Arabia 7: 303-317.

References [18] and [19] should also be mentioned in the Introduction.

Methods:

P. 3: It should be mentioned that the two secondary freshwater fish species are Aphaniops dispar and A. stoliczkanus. Further below, Planiliza abu (“Mugil cephalus”) is mentioned as a secondary freshwater fish, which is not correct.

The authors state that 16 sites were sampled, with reference to the supporting information provided. However, the supporting information only consists of maps and a table with geographic coordinates without further explanations. The authors should explain what the areas highlighted in red and the black dots mean and how this information was derived.

Supporting Information: The first map attributed to Mugil cephalus obviously shows the Al-Ahsa record of Planiliza abu. What does the area highlighted in red refer to?

For the 16 sites sampled, more detailed information needs to be provided. Were fishes collected, and if so, how? By electric fishing gear, nets and/or visual observations? Has any quantitative method been applied, have specimens been counted by species, size classes, etc.? Have voucher specimens been deposited and, if so, where? Who has done the sampling and how long did the person(s) spend at each sampling site? When were the sites sampled? Dates are important, because wadi fish are usually r-strategists with a very pronounced seasonality in population size and distribution.

“Major drainage systems and wadis in Saudi Arabia” should go into the Introduction rather than the Results section.

P. 3: “Two main natural drainage systems are found in Saudi Arabia”: It should be explained that these drainage basins are in western Saudi Arabia. There are also freshwater bodies in the Central Region and the Eastern Province, but only one species of secondary freshwater fish occurs there. It should be explained that eastern drainage systems end in an endorheic basin.

Caption of Fig. 1: It should be explained what G1 – G4 stands for, which could be done with reference to the text (“groups G1 to G4 explained in the text”). Only M refers to the Al-Madinah Region. Rather than writing that the western wadis drain into the Red Sea, it should be stated that they drain towards the Red Sea (most of them reach the Red Sea only seasonally or not at all, as mentioned further below in the text).

P. 3: The Sarawat Mountains are mentioned in the text with reference to Fig. 1. However, the Sarawat Mountains are neither mentioned on the map nor in the caption.

Throughout the manuscript: The spelling of geographical names must be consistent, e.g. “Wadi Turabah” vs. “Wadi Turbah”.

P. 4: “Krupp [1982]“ should read “Krupp [1983]”

P. 4: A wadi is a river bed, rather than a river valley, usually with a seasonal and/or intermittent water bodies.

It should be mentioned that relatively high discharge in the southern wadis is due to the monsoonal regime in that part of Saudi Arabia.

P. 4: “possibly that may indicating the wadis in this group” should read “possibly indicating that wadis in this group”

P 4 and caption of Fig. 1: Al-Madinah or Al-Madinah region should read “Al-Madinah Province” (to distinguish it from the city of Al-Madina Al-Munawwara).

P. 4: The sentence “The main wadi channel in the Al-Madinah, Wadi Al-Hemad or Idham” is incomplete, please check.

P. 5 and Abstract: The term “Biogeographical distribution” does not make sense and needs reconsideration. It should be replaced by “geographic distribution”.

P. 5 “The distribution of indigenous and secondary freshwater fish species” should read “The distribution of indigenous primary and secondary freshwater fish species” (secondary freshwater fish species are also indigenous).

P. 5: The heat map is a welcome approach, but more detailed information needs to be provided on how the “density of each fish species group” was calculated, given the fact that many of the published records are very old, published more than 45 years ago, while the samples on which some of these records are based are even older. As stated in the paper, water bodies, ecological conditions, etc., have changed considerably over time and there are many threats that have changed population densities. In many cases, species that have been recorded in the past are likely to be no longer there.

P. 5: The statement “Secondary freshwater fish species include Aphaniops stoliczkanus (Day) and Mugil cephalus.“ needs reconsideration (see comments here above).

P. 5: “biological control” should read “vector control” or “mosquito control”.

P 5: “species such Carassius auratus” should read “species such as Carassius auratus”.

Throughout the paper: The use of genera spelled out vs. abbreviated is inconsistent and needs attention, e.g. on P. 5 “Garra tibanica, Garra sahilia, C. acinaces, A. arabicus, Acanthobrama hadiyahensis” should read “Garra tibanica, G. sahilia, Cyprinion acinaces, Arabibarbus arabicus, Acanthobrama hadiyahensis” (spell out the genus when first mentioned, followed by an abbreviation, if mentioned again in the same paragraph).

Species authors should only be given once upon first mention of a species.

P. 6: “Acanthobrama hadiyahensis is endemic to certain wadis” should read “Acanthobrama hadiyahensis is only known from very few water locations”. All of these locations are within the same wadi system. The supplementary map shows 4 locations, 3 of which are mentioned in the text; better list all 4.

P.6: “Aphaniops dispar and Aphaniops stoliczkanusis are land-locked marine species” is incorrect; they are both secondary freshwater fishes (which also occur in the sea and other saline water bodies). Better reword: “Aphaniops dispar is a secondary freshwater fish occurring in the western drainages, while A. stoliczkanus is known from the Central Region and the Eastern Province”.

P 6: “Ctenopharyngodon Idella” should read “Ctenopharyngodon idella”.

P. 6: I assume that the record of Garra buettikeri from Wadi Al-Bagarah by Hamidan & Shobrak (2019) is a misidentified G. tibanica or G. sahilia.

P. 6 and 7: Aphanius dispar and A. stoliczkanus are not land-locked fish species (see comments here above).

Throughout: Abbreviations used in the text, such as EP, MR, MRN, need to be explained upon first mention in the text, or a List of Abbreviations should be added.

Fig. 5: “Carassius Carassius” should read “Carassius carassius”.

Fig. 5: Records of non-native freshwater species are incomplete, Oreochromis mossambicus is missing entirely.

P. 7 “tilapia species” should read “tilapiine cichlids”.

P. 8: “A. hadiyahensis [Critical]” should read “A. hadiyahensis [Critically Endangered]”.

P. 8: “are declining” should read “are declining in population size”.

Discussions: Many statements from the Results section are repeated here without being discussed any further. These statements should be removed. The Discussion requires re-structuring. It may be condensed considerably without losing substance. Some background information included in the Discussion should be moved to the Introduction.

P. 8: The statement “The distribution of the indigenous freshwater fish fauna appears to be

driven by biogeographical features” does not make sense and needs to be reconsidered. Distribution is always a result of paleogeographic events and ecological conditions.

P. 8: “environmental differences between wadis” should read “physical and ecological differences among wadis”.

P. 8: “Environmental conditions, especially flow regime …” should read “Differences in environmental conditions, especially flow regime …”.

P. 9: “and have invaded by fish” should read “and have been invaded by fish”.

Throughout the paper: Mugil cephalus should be replaced by Planiliza abu and Aphaniops stoliczkanus is not a marine species.

P. 9: The context of the statement “but these fish are also heavily exploited” is unclear. This probably refers to Ctenopharyngodon idella, but certainly not to Aphaniops stoliczkanus.

Information about dams in Saudi Arabia given in the discussion is very important. However, this is background information that needs to go into the Introduction.

The statement “These could be a major issue for the sustainability of these species because the dams …” should be more affirmative, these are major threats. Same for the statement “It appears that fish survive dry and low flow conditions …”, which is a well-known fact.

The Discussion identifies some needs for further studies in the future, which is very important and more requirements and opportunities for further research should be outlined. The effects of climate change need more consideration.

P. 11: 2nd sentence of the 2nd paragraph repeats statements made further above in the Discussion and should be deleted.

References:

Throughout the list of References, the use Italics is inconsistent and needs attention.

P. 13: “Al-Ghamdi, H. S., & Abu-Zinadah, O. (1998)” should read “Al-Ghamdi, H. S., & Abu-Zinadah, O.A. (1998).”

P. 13: “Borkenhagen, K. & Krupp, F. (2013). Taxonomic revision of the genus Carasobarbus

karaman, 1971 …“ should read “ Borkenhagen, K. & Krupp, F. (2013). Taxonomic revision of the genus Carasobarbus Karaman, 1971 …” (Karaman is the author of the genus).

The correct and complete citation of [4] is: Coad, B.W., Alkahem, H.F. & Behnke, R.J. 1983. Acanthobrama hadiyahensis, a new species of cyprinid fish from Saudi Arabia. National Museums of Natural Sciences, Publications in Natural Sciences, 2, i–v + 1–6.

Supplementary Table S1. (The locations of water bodies): This needs more explanations. Are these the records from the literature and the sampling sites? If so, the sources (References) and the dates of the visits to the sampling sites should be added. Abbreviations need to be explained.

6. PLOS authors have the option to publish the peer review history of their article (what does this mean?). If published, this will include your full peer review and any attached files.

Reviewer #1: **Yes: **Jörg Freyhof

Reviewer #2: **Yes: **Friedhelm Krupp

---

## [Author Response · Author response to Decision Letter 0]

10 Sep 2024

All reviewers comments addressed in the response to reviewers document attached

---

## [Decision Letter · Decision Letter 1]

25 Sep 2024

Geographical distribution of freshwater fishes in Saudi Arabia

PONE-D-24-14979R1

Dear Dr. alharthi,

We’re pleased to inform you that your manuscript has been judged scientifically suitable for publication and will be formally accepted for publication once it meets all outstanding technical requirements.

Kind regards,

Benigno Elvira, Ph.D.

Academic Editor

PLOS ONE

Additional Editor Comments:

Some minor remarks for the authors’ consideration:

Depository of samples: Data based on the first author’s collections are only objectively verifiable, if specimens have been permanently deposited in a scientific collection. The authors should state whether this has been done and, if so, which collection.

If specimens have been deposited in one or more scientific collections, have they been transferred to 70% ethanol? Formaldehyde is not suitable for permanent collections.

The authors state: “we agree that Aphaniops stoliczkanus and Mugil cephalus do not fit the classification of secondary freshwater species”. However, A. stoliczkanus is a secondary freshwater fish. Please, correct it.

Reviewers' comments:

All comments have been addressed

2. Is the manuscript technically sound, and do the data support the conclusions?

Reviewer #2: Yes

3. Has the statistical analysis been performed appropriately and rigorously? 

Reviewer #2: Yes

4. Have the authors made all data underlying the findings in their manuscript fully available?

Reviewer #2: Yes

5. Is the manuscript presented in an intelligible fashion and written in standard English?

Reviewer #2: Yes

7. PLOS authors have the option to publish the peer review history of their article (what does this mean?). If published, this will include your full peer review and any attached files.

Reviewer #2: **Yes: **Friedhelm Krupp

---

## [Editor Report · Acceptance letter]

13 Nov 2024

PONE-D-24-14979R1 

PLOS ONE

Dear Dr. Alharthi, 

I'm pleased to inform you that your manuscript has been deemed suitable for publication in PLOS ONE. Congratulations! Your manuscript is now being handed over to our production team.

Kind regards, 

on behalf of

Prof. Benigno Elvira 

Academic Editor

PLOS ONE